# The Importance of Monitoring and Factors That May Influence Leg Length Difference in Developmental Dysplasia of the Hip

**DOI:** 10.3390/children9121945

**Published:** 2022-12-12

**Authors:** Rajiv M. Merchant, Jaap J. Tolk, Anouska A. Ayub, Deborah M. Eastwood, Aresh Hashemi-Nejad

**Affiliations:** 1Norfolk and Norwich University Hospital, Norwich NR4 7UY, UK; 2Erasmus MC Sophia Children’s Hospital, 3015 CN Rotterdam, The Netherlands; 3The Royal London Hospital, London E1 1FR, UK; 4The Royal National Orthopaedic Hospital, Stanmore, Middlesex HA7 4LP, UK

**Keywords:** developmental dysplasia of the hip, limb length discrepancy, avascular necrosis

## Abstract

In unilateral Developmental Dysplasia of the Hip (DDH), avascular necrosis (AVN), femoral or pelvic osteotomy, and residual dysplasia causing subluxation of the proximal femur may influence Leg Length Discrepancy (LLD). This can lead to gait compensation, pelvic obliquity, and spinal curvature. The aim of this study is to determine the prevalence of LLD, establish which limb segment contributes to the discrepancy, describe how AVN influences LLD, and ascertain variables that may influence the need for LLD corrective procedures. Methodology: This study assessed long-leg radiographs at skeletal maturity. Radiographs were assessed for the articulo-trochanteric distance (ATD) and femoral and tibial length. AVN was classified according to Kalamchi–MacEwen. Results: 109 patients were included. The affected/DDH leg was longer in 72/109 (66%) patients. The length difference was mainly in the subtrochanteric segment of the femur. AVN negatively influenced leg length. Older (≥three years) patients with multiple procedures were more likely to have AVN. LLD interventions were performed in 30 (27.5%) patients. AVN grade or type of DDH surgery did not influence the odds of needing a procedure to correct LLD. Conclusions: Procedures to correct LLD were performed irrespective of previous DDH surgery or AVN grades. In most patients, the affected/DDH leg was longer, mainly in the subtrochanteric segment of the femur, largely influenced by femoral osteotomy in patients with multiple operative procedures for DDH. We recommend careful monitoring of LLD in DDH.

## 1. Introduction

Management of developmental dysplasia of the hip (DDH) aims at establishing a stable, concentrically reduced hip that allows for normal remodelling of the acetabulum and femoral head [1,2]. This can be achieved by combining operative and non-operative modalities depending on age and severity of presentation [3,4,5,6]. While the focus of the clinician is primarily on the hip, leg length discrepancies can arise and should be appropriately monitored and treated where necessary [7,8].

Leg length discrepancy (LLD) has been found to be more common in the presence of DDH than in controls [9]. In unilateral DDH, an LLD may arise due to avascular necrosis (AVN), osteotomies of the femur, pelvis, or subluxation of the proximal femur caused by residual dysplasia [10]. The ipsilateral leg is usually longer and often attributed to femoral overgrowth after osteotomy or excessive growth driven by dysplasia (in the absence of a femoral osteotomy) [7,8,9]. History of a femoral osteotomy has been identified as an independent risk factor for ipsilateral limb overgrowth, despite the initial loss of length after a varus osteotomy was performed [7,11].

LLD in patients with unilateral DDH can be problematic for both patients and surgeons. It results in a pelvic tilt leading to gait asymmetry, spinal scoliosis, flexion contracture knee (in the longer leg), or equinus contracture at the ankle (in the shorter leg) [12,13,14,15,16,17]. Surgical considerations include the timing of the guided growth interventions before skeletal maturity, the need for limb elongation procedures after skeletal maturity when the difference is large, and appropriate planning to ensure equal length after hip arthroplasty [8,13,18,19,20,21].

Leg length differences have previously been studied in patients with DDH. Yoon et al. [7] estimated length differences in 101 children with DDH, estimating the difference on standing ap pelvis radiographs. They reported limb overgrowth as common and mainly related to femoral osteotomy. Zhang et al. [20] studied long-leg radiographs of 67 skeletally mature patients with unilateral developmental hip dislocations and found that tibial length, lesser trochanter to tibial plafond length, and overall leg lengths on the affected leg were significantly longer, regardless of high or low dislocations. LLD is important to quantify since Tolk et al. [22] demonstrated a trend towards impaired acetabular development in unoperated DDH patients with greater limb length discrepancy. Other studies have shown an increased risk of total hip replacement on the longer side [23].

The aim of this study is to identify the prevalence of LLD, quantify which segment of the limb contributes to the discrepancy, and describe the influence of AVN on LLD in patients with unilateral DDH. This study also tries to ascertain variables that may influence the need for LLD interventions. This would forewarn clinicians and management and provide a better follow-up guide.

## 2. Materials and Methods

This study reports on a consecutive retrospective case series of all patients treated for DDH between January 2008 and December 2020 in our institution. Inclusion criteria were patients with unilateral DDH referred to our unit for managing primary or residual DDH prior to skeletal maturity with the availability of adequate long-leg radiographs. Exclusion criteria were patients with: (1) associated pathology affecting leg length (e.g., Neuromuscular conditions, congenital abnormalities, or skeletal dysplasia), (2) open triradiate cartilage at the time of review, and (3) bilateral cases. Patients with successful Pavlik harness treatment were not followed up after age of 5 unless they required further intervention, as per our published protocol and, therefore, were excluded from our cohort [23]. No ethical approval was required as this study was classed as an audit of historically treated patients.

### 2.1. Data Collection

Electronic patient files were reviewed for patient and treatment characteristics. Factors recorded were age at diagnosis, age at a final follow-up, side effects, and treatment. Along the course of treatment, some patients required multiple procedures. To reduce this ambiguity, patients were categorised into groups based on the last successful procedure. Groups included patients with the following results:(1)Successful closed reduction;(2)Failed (or successful) closed reduction proceeding to an open reduction without bone surgery;(3)Reduction proceeding to a femoral osteotomy;(4)Reduction proceeding to a pelvic osteotomy;(5)Reduction proceeding to a femoral and pelvic osteotomy.

### 2.2. Radiographic Measurements

Our follow-up protocol recommends a clinical and radiological review, including leg-length assessments at prescribed time points [24].

Radiographic measurements were performed on the calibrated, standardised long-leg standing radiographs. Adequate radiographs were defined as patellae positioned forward; bony landmarks were visible with the presence of a templating ball or scale measure. All measurements were performed using TraumaCad (Brainlab, Petach-Tikva, Israel) software (see Figure 1) [25]. Measurements included the articulo-trochanteric distance (ATD), femoral length, and tibial length. Measurements were performed using measurement tools in the software. ATD was measured by placing markers on the tip of the greater trochanter and superior aspect of the femoral head on a line oriented along the axis of the femur (Figure 1a). Femoral and tibial lengths were calculated by the software after the appropriate identification of bony landmarks by the investigator (Figure 1b,e). Proximal femoral growth disturbance was classified according to Kalamchi–MacEwen [26] for a grade of avascular necrosis. All three authors independently reviewed the measurements, and discrepancies were resolved with consensus [27].

### 2.3. Statistical Analysis

For descriptive statistics of continuous variables, means were reported, and for discrete variables, counts and percentages were presented. Mean leg length differences between the longer leg vs. the shorter leg and DDH affected vs. unaffected legs were compared using *t*-test samples for each of the leg segments measured (total leg length, tibial length, total femoral length, ATD, and subtrochanteric femoral length) and reported as mean differences with 95% confidence intervals.

An analysis was made to assess whether having multiple procedures or procedures at a younger age (≤3 years) influenced the final radiological outcome according to AVN graded by Kalamchi–MacEwen [26]. The statistical significance was calculated using chi-square and Mann–Whitney U test. An association between the various treatment modalities and the need for LLD intervention was also analysed, and the results were presented as odd ratios with 95% confidence intervals. A similar analysis was performed to assess the relationship between AVN grade and the need for LLD intervention.

All data were tabulated, and SPSS (IBM Corp. Released 2020. IBM SPSS Statistics for Macintosh, Version 27.0.) was used for statistical analysis.

## 3. Results

During the study period, 248 DDH patients were identified. 109 patients met all the criteria for the follow-up (Figure 2). Table 1 describes the patient characteristics and frequency of procedure types.

### 3.1. Avascular Necrosis

The AVN rate (type II, III, IV) was 52% (56/109). The most common grade of AVN encountered was Type II (Table 2).

AVN rates were higher in patients that had multiple procedures (Figure 3), with the difference being statistically significant (*p* < 0.001 chi-square). There were fewer cases of ‘poor’ AVN (type III and IV) (*p* = 0.029, chi-square) in patients who had surgery at an earlier (≤ three years) age (Table 2). The grade of AVN affected the ipsilateral leg length negatively, i.e., the ipsilateral leg transitioned from being longer to shorter with increasing AVN grade (Figure 4).

### 3.2. Leg Length and Segmental Differences

At skeletal maturity, the ipsilateral leg was longer than the unaffected leg in 72 (66%) patients and shorter in 34 (32%) patients. Three patients had symmetric leg lengths at the final follow-up. The difference was of no clinical significance (i.e., difference <1 cm) in 70/109 (64.2%) patients, leaving 36/109 (33%) patients with a significant difference of ≥1 cm.

When the ipsilateral leg was longer, most of the segmental difference was in the subtrochanteric length. Conversely, when it was shorter, most of the segmental difference was in the articulo-trochanteric distance, with a statistically significant difference (Table 3).

### 3.3. Interventions for Leg Length Difference

In total, 30 (27.5%) patients underwent n LLD intervention. Of those, 9 (30%) patients had a residual leg length inequality of ≥1 cm after an LLD intervention. This latter group comprised 3 patients, each with no AVN, type II and III AVN. Surgically, the same group included 4/9 patients with revision femoral osteotomies; 2/9 had closed reduction only, and 3/9 had an open reduction, pelvic and femoral osteotomies.

Corrective procedures for LLD prior to skeletal maturity included 12 patients who underwent a permanent drill epiphysiodesis and 14 patients who underwent a temporary epiphysiodesis with medial and lateral tension band plates. One eleven-year-old patient had surgery for the hexapod external fixator frame for a 6 cm leg length difference, resulting from multiple DDH surgeries. Three patients underwent acute shortening of the longer leg after skeletal maturity. The majority (22/30) of procedures were carried out to shorten the DDH-affected limb. Of the remaining eight patients whose ipsilateral legs were lengthened, 4/8 patients had type IV AVN, 3/8 had type III AVN, and one patient had no AVN.

The odds of needing an LLD intervention for each surgical treatment group and AVN grade were calculated (Table 4 and Table 5). The risk was greater for patients with a history of femoral osteotomy, combined pelvic and femoral osteotomy (not pelvic osteotomy alone), groups with multiple prior operations, and type IV AVN groups. However, the confidence interval in all groups overlapped the null value; therefore, no statistically significant conclusions could be drawn. LLD interventions were required across all surgical groups and AVN grades.

## 4. Discussion

This study shows that the leg length discrepancy is common in patients with unilateral DDH, with nearly twice the number of patients with a residual LLD ≥1 cm at skeletal maturity compared to the general population despite appropriate surveillance and treatment [28]. Leg length discrepancy can arise and would need treatment irrespective of the type of procedure or AVN grade. Higher AVN grades were more common in patients with multiple procedures and surgery performed after three years of age and negatively impacted the length of the ipsilateral leg. Conversely, in patients with low-grade or no AVN, the ipsilateral leg was, on average, longer than the unaffected leg.

### 4.1. Leg Length Discrepancy in DDH

The ipsilateral leg was longer than the unaffected leg in seventy-two (66%) patients, similar to other reports of 66-78% in the literature [8,20]. When considering an LLD of ≥1 cm, the prevalence in the general population is 15% [28]. This study identified 36 (33%) patients with a residual LLD of ≥1 cm after surveillance and treatment. This value is lower than that reported by Yoon et al. [7], who identified 44% of their 105 patients with a ≥1 cm LLD. Therefore, appropriate surveillance and timely interventions for LLD are recommended in DDH patients.

Unsurprisingly, this study identified the femur to significantly contribute to LLD with little to no contribution from the tibia. When the DDH leg was shorter, it was the ATD that contributed to the discrepancy. As the ATD is influenced by the severity of AVN, this explains why the DDH leg was, on average, shorter with worsening AVN grade. Femoral overgrowth and its influence on length after the femoral osteotomy is well documented [7,8,9]. Zhang et al. [20] reviewed 67 patients with a mean age of 25 years, assessing segmental leg lengths in Hartofilakidis’ low and high dislocations of patients with DDH (excluding patients with femoral osteotomy). They reported the ipsilateral femoral shaft to be longer in 78% of cases, regardless of high or low dislocation. If we take this to be the natural history of femoral overgrowth with the femur untreated, this will justify our previous recommendation to shorten the femur at the time of osteotomy. This reduces the soft tissue tension protecting the femoral head from AVN and pre-empting the sequelae of overgrowth [11]. LLD in different lower limb segments is particularly important during hip arthroplasty as surgeons may focus on restoring the hip centre of rotation and not consider the difference in diaphyseal lengths, resulting in a length discrepancy [27].

### 4.2. Consequences of LLD in DDH

LLD can lead to a gait asymmetry, while there are no measurable kinematic changes in minor differences ≤1 cm, a pelvic obliquity compensation can arise above a 2 cm difference [12]. In the coronal plane, pelvic obliquity can cause dynamic acetabular dysplasia in the longer leg or over-coverage and potential impingement of the shorter leg, both of which have been shown to affect the long-term outcome of hip joint development [13]. Segmental LLD in DDH has implications on hip arthroplasty planning as subtrochanteric length differences can be easily missed [29]. The longer leg tends to do more mechanical work, and energy expenditure increases with increasing leg length discrepancy [30,31]. It is, therefore, essential to monitor leg length discrepancy in DDH patients and consider treatment where necessary to limit the impact on gait dysfunction.

### 4.3. Appropriate Monitoring of LLD in DDH

The literature suggests that clinical examination and tape measures are effective screening tools. However, imaging modalities are more accurate in measuring leg length differences [32]. Our protocol of long-leg standing radiographic follow-up of DDH from age of five allows early diagnosis and planning of appropriate intervention for the management of the leg length discrepancy [24]. Early detection should be managed with simple raises, and follow-up allows for surgical intervention based on bone age. A moderate LLD found in these patients can be treated safely and reliably with epiphysiodesis. The success rate in our case was nearly one in three patients [10,33].

The appropriate imaging modality for monitoring LLD should be chosen, taking into account the radiation exposure, patient movement artefact, and beam distortion [32]. One option for such monitoring is using standing anteroposterior pelvis radiographs and using femoral head height difference (FHHD) to assess leg lengths [7]. This modality does not allow asymmetries resulting from apparent limb length discrepancy due to adduction, abduction, and flexion contractures or fixed spinal deformities. We agree with the opinion of Zhang et al. [20] that using the lesser trochanter on standing AP pelvic radiographs to predict LLD is unreliable. The use of full-length standing anteroposterior radiographs for preoperative templating is advisable for this special group of patients, and its use in our follow-up protocol is justified [24].

### 4.4. Treatment of LLD in DDH

There is a paucity of literature on whether interventions are needed to address leg length discrepancies in DDH patients. This study found that 30 (27.5%) patients required an LLD intervention, similar to Yoon et al. [7], who reported 23.7%. Inan et al. [10] reported that 12 of their 398 patients required epiphysiodesis, but they did not report on other modalities used to correct LLD or leg length differences within the cohort of patients. In our study, LLD interventions were required across all procedural groups, and we were unable to show a strong correlation with any procedure (Table 4). There was a trend towards increased odds of the need for intervention in the femoral osteotomy group; however, this was not statistically significant. We attribute this to several factors, which include small numbers in each group, successful surveillance and routine shortening of the femur performed during osteotomy at our institution.

### 4.5. Influence of AVN on LLD in DDH

Fifty-two per cent of our patients were identified to have type II or higher grade of post-avascular necrosis-related proximal femoral growth disturbance (Table 2). More severe grades (III and IV) were more prevalent in the groups with multiple prior operations and older age groups (Table 2 and Figure 3). LLD Interventions were required across all AVN grades, and Type IV had the highest odds of requiring an intervention; however, there were only four patients in the subgroup (Table 5). All LLD procedures in patients with Type III and IV AVN relatively lengthened the shorter ipsilateral leg. Inan et al. [10] found AVN to be a common risk factor in their series of 12 patients that required epiphysiodesis. All their patients had type II or higher AVN. In our series, 15 (50%) patients that required a limb length discrepancy intervention had type II or higher-grade AVN. However, we were not able to find any statistical significance (Table 5).

### 4.6. Consequences of Untreated Leg Length Discrepancy

Untreated leg length discrepancy can cause pelvic obliquity, resulting in compensatory spinal curvature, also called functional scoliosis [34]. This can result in the wear of facet joints over a long period and a more structural or fixed curve [13,34]. There is currently no evidence on how much LLD contributes to developing a spinal deformity [13,34]. Other compensatory mechanisms include increased knee flexion, external ankle rotation on the longer side, and ankle equinus on the shorter side [13]. These can potentially transition from flexible gait adaptations to fixed contractures if LLD is left untreated [13].

The strengths of this study include a cohort of patients who were systematically followed up by an established protocol. To the best of our knowledge, this is the largest cohort of patients with DDH, followed up by standing radiographs. All radiographs were standardised and calibrated. Limitations include the retrospective nature of the study and the fact that we offered LLD intervention based on clinical necessity and patient choice. This may underestimate the true impact of the leg length discrepancy on our patients.

## 5. Conclusions

The present study showed that LLD is higher in patients treated for DDH than in the normal population. This knowledge led to the need for a leg length equalisation procedure in 27.5% of the patients. In most patients, the DDH leg was longer, mainly arising from the subtrochanteric region, with data showing a trend towards this femoral overgrowth after femoral osteotomy and in patients with multiple operative procedures to the affected hip. In patients with shorter DDH legs, this was associated with the occurrence of AVN. Our results underline the need for careful monitoring of LLD with long-leg standing films at appropriate stages in patients treated for DDH.

## Figures and Tables

**Figure 1 children-09-01945-f001:**
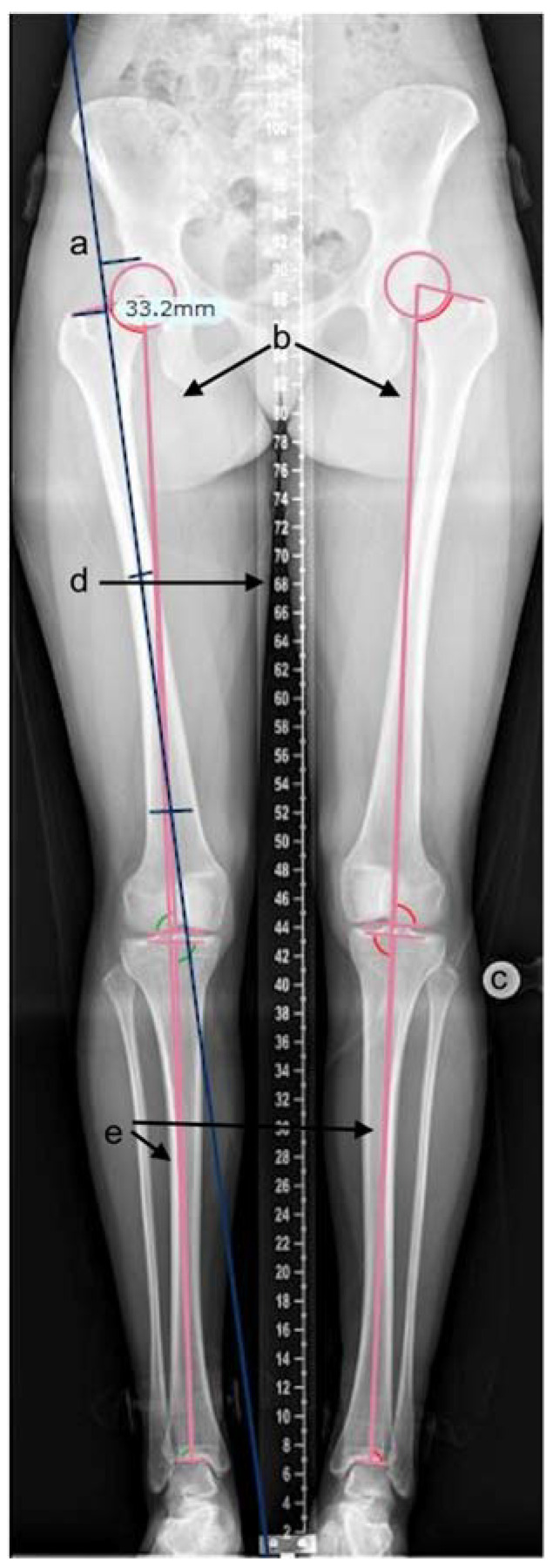
TraumaCad measurement of leg length parameters. (a) tool for articulo-trochanteric distance, (b) femoral length tool, (c) templating ball for calibration, (d) measuring ruler, (e) tibial length tool.

**Figure 2 children-09-01945-f002:**
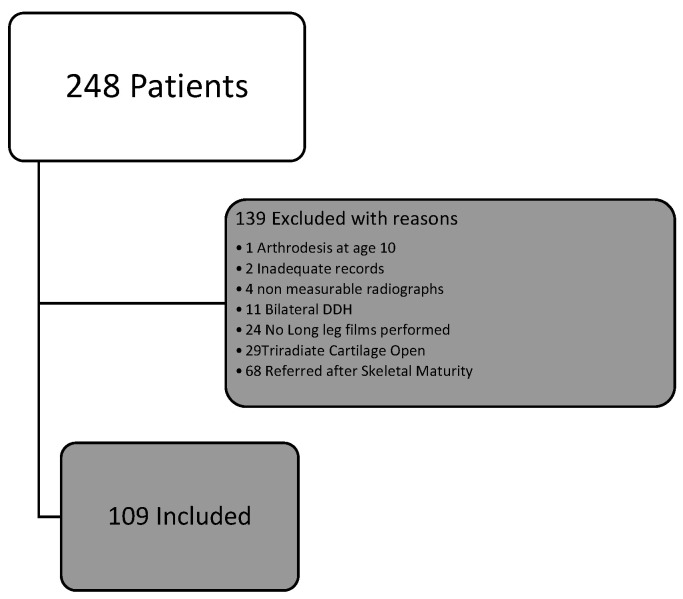
Patient inclusion and exclusion flow-chart.

**Figure 3 children-09-01945-f003:**
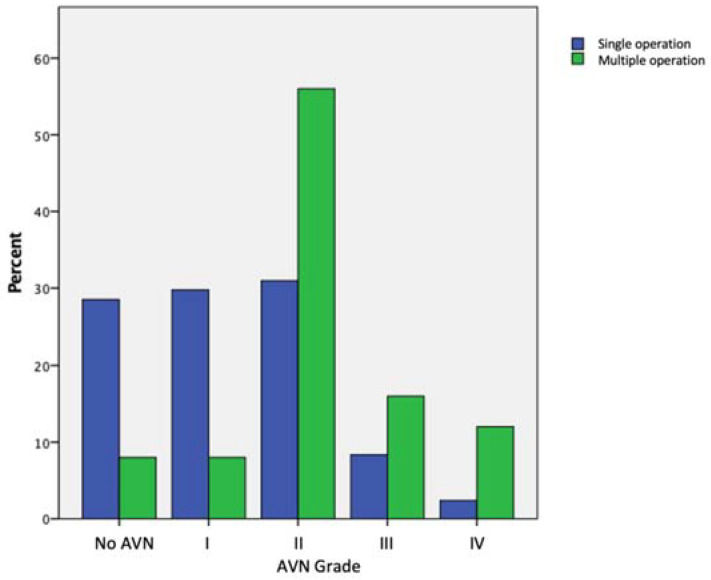
AVN grade frequency distribution for patients with single and multiple procedures.

**Figure 4 children-09-01945-f004:**
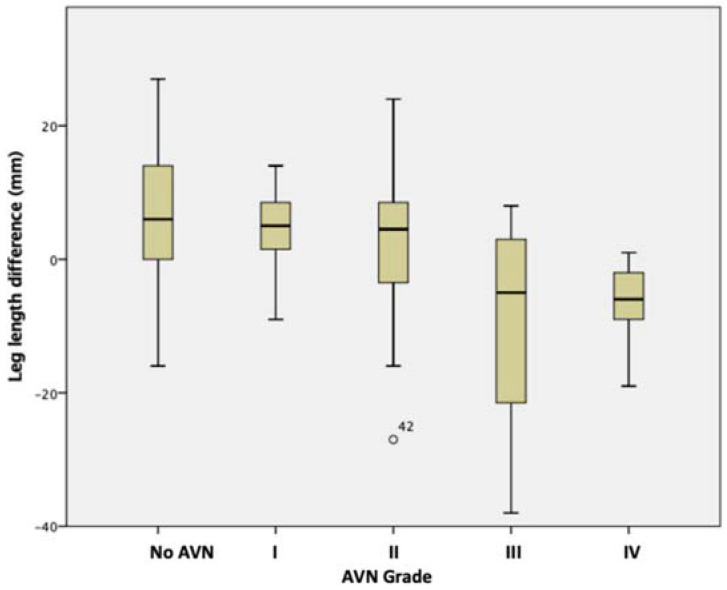
The average leg length difference between the DDH leg and the unaffected leg, grouped by AVN grade.

**Table 1 children-09-01945-t001:** Patient demographics and frequency procedure types.

Patient Characteristics	*n* = 109
Mean age at referral with range in years	2.7 (0–13)
Mean age at diagnosis with range in years	1.5 (0–10.5)
Mean age at follow-up with range in years	15.2 (10.6 to 49.9)
Side affected	Right 51/Left 58
Sex	Female 95/male 14
**Surgical procedure Groups (Last surgical Procedure)**	***n* = 109**
Closed reduction	17 (15.8%)
Open reduction only	17(15.8%)
Reduction + Femoral osteotomy	30 (27.5%)
Reduction + Pelvic osteotomy	15 (13.6%)
Reduction + Pelvic and femoral	30 (27.3%)
Total	109(100%)
Multiple Surgeries (bone and/or soft tissue)	25/109 (23%)
Values presented as mean with standard deviation between brackets for continuous variables and count with percentages between brackets for dichotomous variables.	

**Table 2 children-09-01945-t002:** AVN distribution in patients who had their first operation at less than or equal to 3 years of age and greater than 3 years of age.

	Age of First Surgery		Total
	≤3 years	>3 years	
No AVN	26/95 (27.4%)	0	26 (23.9%)
AVN Type I	24/95 (25.3%)	3/14 (21.4%)	27 (24.8%)
AVN Type II	32/95 (33.7%)	8/14 (57.1%)	40 (36.7%)
AVN Type III	9/95 (9.5%)	2/14 (14.3%)	11 (10.1%)
AVN Type IV	4/95 (4.2%)	1/14 (7.1%)	5 (4.6%)
Total	95 (100.0%)	14 (100.0%)	109 (100.0%)

**Table 3 children-09-01945-t003:** Total leg length and segment lengths represented with median values and range.

Segmental Leg Length Difference	DDH Leg Longer (Mean Length Difference in mm, CI) (*n* = 72)		DDH Leg Shorter (Mean Length Difference in mm, CI) (*n* = 34)	
**Total leg length discrepancy**	8.25 (6.8, 9.6)	p < 0.001	−9.5 (−12, −6.4)	p < 0.001
**Femoral length difference**	7.5 (6.04, 8.8)	p < 0.001	−6.5 (−9.6, −3.2)	p < 0.001
**Tibial difference in**	0.5 (−0.5, 1.4)	p = 0.3	−1.9 (−4.2, 0.3)	p = 0.09
**Articulo-trochanteric distance difference**	0.28 (−0.9, 1.4))	p = 0.63	−5.2 (−9.2, −1.01)	p = 0.016
**Diaphyseal length difference**	7.1 (5.4, 8.9)	p < 0.001	−1.32 (−5.9, 3.3)	p = 0.565

**Table 4 children-09-01945-t004:** The risk of the leg length corrective intervention stratified for the treatment group.

	Frequency (*n* [%])	OR	95% CI
Closed reduction (*n* = 17)	4/17 (23.5%)	0.78	(0.23–2.62)
Open reduction (*n* = 18)	2/18 (11.1%)	0.31	(0.07–1.42)
Femoral osteotomy (*n* = 30)	11/30 (36.7%)	1.83	(0.74–4.52)
Pelvic osteotomy (*n* = 15)	4/15 (26.7%)	0.95	(0.28–3.26)
Pelvic and femoral (*n* = 29)	9/29 (31.0%)	1.18	(0.47–2.99)
Multiple operations (*n* = 25)	9/25 (36.0%)	1.69	(0.65–4.38)

OR; odds ratio; CI; confidence interval.

**Table 5 children-09-01945-t005:** The risk of leg length corrective intervention, stratified for the AVN grade.

	Frequency (f/n [%])	OR	95% CI
No AVN/Type I (*n* = 53)	15/53 (28.3%)	1.08	(0.47–2.50)
Type II (*n* = 40)	8/40 (20.0 %)	0.53	(0.21–1.35)
Type III (*n* = 11)	3/11 (27.3%)	0.99	(0.24–4.00)
Type IV (*n* = 5)	4/5 (80%)	12.00	(1.28–112.25)

Risks stratified for avascular necrosis (AVN) grade according to Kalamchi–McEwan. OR; odds ratio; CI; confidence interval.

## Data Availability

Not applicable.

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
