# Peer review of "The Importance of Monitoring and Factors That May Influence Leg Length Difference in Developmental Dysplasia of the Hip"

_children, 2022, doi:10.3390/children9121945_

Round 1

Reviewer 1 Report

An article on an interesting and so far poorly described topic - Leg length discrepancy (LLD) in unilateral Developmental Dysplasia of the Hip.

I have a few comments:

Introduction:

Line 38: please correct the citation - should be [9] is:… .. controls.9

Line 46-49: please add information that unbalanced limb shortening may cause scoliosis and faster development of degenerative changes and compensation mechanisms such as: knee contracture in the longer leg and foot equinus of the shorter leg.

Line 49-52: Limb elongation surgery may also be necessary in growing children if the shortening is large.

In Introduction, please write if there were any thematically related articles - evaluating Leg length discrepancy (LLD) in unilateral Developmental Dysplasia of the Hip.

Materials and methods:

Please describe exactly how it was measured: the Articulo-trochanteric distance (ATD), femoral length and tibial length.

The standard length of the lower limb is measured from the upper edge of the pelvis to the upper edge of the talus - why did you not measure it like this?

The length of the pelvis also affects the shortening.

Please describe the criteria for subdividing the AVN.

The picture is of poor quality and the pelvis is not fully captured.

Discussion:

Please add information that unbalanced limb shortening may cause scoliosis and faster development of degenerative changes and compensation mechanisms such as: knee contracture in the longer leg and foot equinus of the shorter leg.

Author Response

Author response

Many thanks for taking the time to review our manuscript

Line 38: please correct the citation - should be [9] is:… .. controls.9

Referencing has been updated

Line 46-49: please add information that unbalanced limb shortening may cause scoliosis and faster development of degenerative changes and compensation mechanisms such as: knee contracture in the longer leg and foot equinus of the shorter leg.

Statement has been altered

Line 49-52: Limb elongation surgery may also be necessary in growing children if the shortening is large.

Statement has been updated

In Introduction, please write if there were any thematically related articles - evaluating Leg length discrepancy (LLD) in unilateral Developmental Dysplasia of the Hip.

New paragraph has been added with thematic articles

Materials and methods:

Please describe exactly how it was measured: the Articulo-trochanteric distance (ATD), femoral length and tibial length.

This information has been included.

The standard length of the lower limb is measured from the upper edge of the pelvis to the upper edge of the talus - why did you not measure it like this?

There were 2 reasons for this

  1. To eliminate human error we relied on software calculation of leg lengths. The software only allowed femoral and tibial bony landmarks to be used.
  2. In patients that had a pelvic osteotomy Pelvic Height can be altered either by type of osteotomy or abnormal apophyseal growth. We decided to avoid this confounder.

Please describe the criteria for subdividing the AVN.

AVN was subdivided according to Kalamchi MacEwen Classification based on radiographic features.

We consider this well documented in the literature and describing in detail in the current paper would be redundant.

The picture is of poor quality and the pelvis is not fully captured.

A new image has been obtained.

Discussion:

Please add information that unbalanced limb shortening may cause scoliosis and faster development of degenerative changes and compensation mechanisms such as: knee contracture in the longer leg and foot equinus of the shorter leg.

This has been updated with an additional paragraph in the Discussion.

Reviewer 2 Report

The topic of the paper is interesting and fits the scope of the journal. The text is relatively well written and composed. I have only minor comments that I believe that help to improve the paper. 

Methods

Please refer if you have ethics committee in research.

Results

Line 113. Please replace “Two hundred and forty eight” with “248”.

Conclusion

Why do you believe that most of patients are female?

Minor comments

Line 38. Please, remove the reference 9 as superscript.

Please check again the manuscript and remove the spaces between words. 

Author Response

Thank you for taking the time to review our manuscript

Methods

Please refer if you have ethics committee in research.

No ethical approval was required as this study was classed as an audit of historically treated patients.(this statement has been included in the methodology)

Results

Line 113. Please replace “Two hundred and forty eight” with “248”.

 This line has been edited as requested

Conclusion

Why do you believe that most of patients are female?

We have not mentioned gender in our conclusion section. However, as DDH is more common in females, this reflects in our patient population.

Minor comments

Line 38. Please, remove the reference 9 as superscript.

This has been changed

Please check again the manuscript and remove the spaces between words. 

Has been edited.

Round 2

Reviewer 2 Report

Thank you for this study. Accepted as it is.